# Effect of Religiosity, Perceived Risk, and Attitude on Tax Compliant Intention Moderated by e-Filing

Mekar Satria Utama *, Umar Nimran, Kadarisman Hidayat and Arik Prasetya

Department of Business Administration, Faculty of Administrative Science, University of Brawijaya, Veteran Street, Lowokwaru, Malang 65145, Indonesia; umar.nimran.ub@gmail.com (U.N.); kadarisman.ub@gmail.com (K.H.); prasetyaarik2@gmail.com (A.P.)
* Correspondence: mekarsatria08.fia@gmail.com; Tel.: +62-341-551611

**Abstract:** This research examined the effect of Religiosity, Perceived Risk, and Attitude on Tax Compliant Intention, moderated by e-Filing. This research used a quantitative approach which involved the Structural Equation Model (SEM). Large taxpayers are generally in the form of agencies and individuals, so the population of this study comprised of companies that are in the Directorate General of Taxation of Large Taxpayers Jakarta, Large Tax Service Offices 1 and 2, totaling 529 companies. Religiosity (X1) and Perceived Risk (X2) significantly influence Attitude (Y1). Furthermore, Attitude (Y1) has a positive and significant effect on Tax Compliant Intention (Y2). e-Filing showed an insignificant moderating effect on the research model. The novelties are the development of the Theory of Planned Behavior by involving other variables that affect taxpayer compliance behavior, namely Religiosity and the perceived risk of taxpayers. In addition, this research involves the e-Filing variable as a moderating variable.

**Keywords:** religiosity; perceived risk; attitude; e-Filing; tax compliant intention

**JEL Classification:** G32; H23; Z12



## 1. Introduction

Predicting individual behavior is known as the Theory of Planned Behavior (TPB), which is an extension of the theory of reasoned action. The concept of intention and behavior, in general, has been studied in Theory of Reasoned Action (TRA), which was first introduced by Fishbein and Ajzen in 1975. In the TRA framework, behavioral intention can largely determine actual behavior. This behavioral intention is an additive function of two variables, namely subjective attitudes and norms. In the TPB model (Ajzen 1991), intention to behave is an intermediate variable in behavior. That is, individual behavior is generally based on the intention to behave. The intention of using technology services is the awareness of the ability to use the services of customers (Nguyen 2020).

Even though the object is the same, not all individuals have the same attitude; it can be influenced by individual circumstances, experiences, information, and the different needs of each individual. Each individual has an attitude that determines their characteristics, one of which is the characteristic of Religiosity (Wibowo and Masitoh 2018; Graafland 2017; Lu et al. 2019). Several recent studies have shown the importance of religious values in human resource management and organizational behavior in an organization. Several studies suggest that organizational change can be carried out through the application of religious values in the workplace (Konz and Ryan 1999; Féry 2003; French and Bell 2001). Religiosity, according to Johnson et al. (2001), is seen as the extent to which individuals are committed to their religion and faith and apply their teachings, so individual attitudes and behaviors reflect this commitment. Mohd Ali (2013) stated that it is hoped that the role of religious values can spur positive behavior and prevent negative behavior towards behavioral obedience to encourage good behavior.

It is believed that to determine the booster of Tax Compliant Behavior, another factor is needed, namely the perceived risk. Based on the Minister of Finance Regulation No. 191/PMK.09/2008 concerning the Implementation of Risk Management in the Ministry of Finance, risk is anything that harms the achievement of objectives measured based on its likelihood and impact. The current condition of Indonesian taxation is required to implement risk management to reduce potential risks to such a low level. Although the final risk is in the form of financial risk, tax risk can be divided into four risks. First, operational risk, which is the risks caused by, among others, the inadequacy and/or malfunction of internal processes, human error, system error, or external problems affecting the operations of the taxpayer. Second, legal risks, which are risks caused by weaknesses in juridical aspects, among others, due to lawsuits from other parties, absence of supporting statutory behavior or engagement weaknesses for a business transaction that has no specificity, clarity, or details regulating the transaction. Third, reputation risk, which is a risk caused by, among others, negative publicity related to business activities or negative perceptions of taxpayers. Fourth, compliance risk, which is a risk caused by taxpayers not complying or implementing the applicable tax laws and regulations.

Electronic tax return, which is implemented by the government, is e-Filing. e-Filing is a method of reporting tax returns that electronically or online through the website of the Directorate General of Taxes (DGT Online) or other official e-Filing channels established by the government. With e-Filing, taxpayers no longer have to go to the tax office to report taxes. Since compliance with the submission of annual tax returns is not 100%, taxes are not yet an effective instrument to realize income redistribution and reduce economic disparities.

Concerning tax compliance, the intention is the intended desire of individual taxpayers to carry out tax compliance or non-compliance behavior. The decision is a personal decision, that is chosen by taxpayers, to obey or disobey tax regulations. Several studies using the Theory of Planned Behavior have shown that intention affects behavior. Research by Bobek and Hatfield (2003) showed that the intention to behave has a positive and significant effect on behavior. The novelties that can be found in this study include the development of the Theory of Planned Behavior from the basic theory proposed by Ajzen (1991) by involving other variables that affect taxpayer compliance behavior, namely Religiosity and the perceived risk of taxpayers.

The findings in this study can provide meaningful theoretical contributions to the development of scientific knowledge. The contribution of this research to the development of science is the research findings on testing and clarification of the theories developed in this study as well as the consistency of findings generated from previous research. The results of this study prove that religiosity is a dominant factor in influencing tax compliant intention. However, carrying out knowledge management practices, is not only formed by this but also requires an adequate perceived risk and attitude. E-Filing, especially Simplicity of the System should be a major concern to further develop knowledge management practices and increase tax compliant intention.

The results of this study are that Religiosity (X1) and Perceived Risk (X2) significantly influence Attitude (Y1). Furthermore, Attitude (Y1) has a positive and significant effect on Tax Compliant Intention (Y2). E-Filing showed an insignificant moderating effect of the research model. The novelties are the development of the Theory of Planned Behavior by involving other variables that affect taxpayer compliance behavior, namely Religiosity and the perceived risk of taxpayers. In addition, this research involves the E-Filing variable as a moderating variable. In the next section, we will discuss in detail some definitions of the variables used, the methods used, a discussion of the results obtained in the study, along with the implications and conclusions of the study.

## 2. Literature Review

### 2.1. Religiosity

Religiosity is defined as how wide the knowledge is, how strong the belief is, how deeply implemented worship and rules are, and how deep the appreciation of the religion

they profess is (Nashori and Mucharam 2007). Religiosity is the strength of an individual's relationship or belief in his religion (Susanti 2016). Religiosity is a complex integration of religious knowledge, feelings, and religious actions within a person. Religiosity, according to Johnson et al. (2016), is seen as the extent to which individuals are committed to their religion and faith and apply their teachings so that individual attitudes and behavior reflect this commitment.

Religiosity, in this study, shows the appreciation and attitude of the life of taxpayers based on the religious values they hold. This appreciation is shown by tax compliance behavior. Religiosity, in general, is associated with cognition (knowledge and religious belief) which affects what is done with emotional attachment or emotional feelings about religion and/or behavior, such as attendance of places of worship, reading holy books, and praying (Elci 2007). According to Glock and Stark (1965), in detail, religiosity has five important dimensions: belief (ideological), religious practice (ritualistic), appreciation, knowledge (intellectual), and practice (consequential).

### 2.2. Perceived Risk

Based on the Minister of Finance Regulation No. 191/PMK.09/2008 concerning the Implementation of Risk Management in the Ministry of Finance, risk is anything that harms the achievement of objectives measured based on likelihood and impact. According to Ilyas and Burton (2013), for potential risks to be reduced, organizations must implement risk management. The current condition of Indonesian taxation requires implementation of risk management to reduce potential risks to a low level. Perception of risk is defined by Oglethorpe and Monroe (1994) as consumer perceptions of uncertainty and possible negative consequences for purchasing a product or service. Perceived risk is the uncertainty of an event that, if it occurs, will cause a loss. According to Nitisusastro (2012), the dimensions of risk perception are Financial Risk, Functional Risk, Physical Risk, Psychological Risk, Social Risk, and Time Risk.

### 2.3. Attitude

The theory of attitudes in this study is a development of the Attitude Toward the Behavior variable in the Theory of Planned Behavior. Attitude toward a behavior is a positive or negative evaluation of an object, person, institution, event, behavior, or intention (Ajzen 2005). In addition, Ajzen (2005) argues that Attitude Toward the Behavior is determined by beliefs about the consequences of behavior or in short, behavioral beliefs. According to Hariyono (2021), the pattern of attitudes, behavior, and local cultural values instilled by parents in children are the basis for developing their future behavior. The definition of attitude was also conveyed by Sarwono and Meinarno (2009) as follows: Attitude is a process of assessment carried out by an individual on an object.

An attitude is a group of beliefs and feelings that are inherent about certain objects and the tendency to act on these objects in certain ways (Calhoun 2006). Attitude is a term that reflects the feeling of pleasure, displeasure, or a neutral feeling of someone about something. Fishbein and Ajzen argued that there are two groups in the formation of attitudes: Behavioral Belief and Evaluation of Behavioral Belief.

Based on several opinions, it can be said that attitude is a reaction or response in the form of an assessment that arises from an individual towards an object. Attitude can also be interpreted as a manifestation of the awareness of the environment. The process that initiates the formation of attitudes is the existence of objects around the individual providing a stimulus that then affects the individual senses; the information captured about the object is then processed in the brain and causes a reaction.

### 2.4. e-Filing

According to Soemitro (2013), tax is the transfer of wealth from the people to the state treasury to finance routine expenses and the surplus is used for public saving which is the main source for financing public investment. Law no. 28 of 2007 states that tax is a

mandatory contribution to the state that is owned by an individual or entity that is coercive based on the law without receiving direct compensation.

Article 6 paragraph (2) of Law no. 16 of 2000 concerning General Provisions and Tax Procedures states that the submission of notification letters can be sent through the Post Office on a registered basis or by other means regulated in a decision of the Director-General of Taxes. From the following statement of the Law, it can be seen that SPT (Tax Return) reporting, in general, has been done so far by submitting directly to the Tax Service Office or sending by a registered mail. With this system, taxpayers must come and meet directly with tax officials. This system also requires a lot of human resources and space and slows down the service due to the manual delivery process. Furthermore, errors in recording occur more easily. Thus, a faster and more accurate administration and service system are needed in the Tax Office.

On 24 January 2005, at the Presidential Office, the President of the Republic of Indonesia together with the Directorate General of Taxes launched the product e-Filing or Electronic Filing System. In simple terms, e-Filing is an implementation of e-Government implementation in tax administration, especially tax return reporting, to support the existing tax system, namely in the form of tax compliance levels, facilitating tax reporting, and reducing tax payment errors.

*2.5. Tax Compliant Intention*

The Tax Compliant Intention Theory in this study is a development of the Intention variable in the Theory of Planned Behavior. The Big Indonesian Dictionary states that Intention is the goal or purpose of an action and the will or desire from the heart to do something. The intention is not always static; intention can change over time. The theory of Reasoned Action (TRA) assumes that humans behave consciously, taking into account the available information, and implicitly and explicitly considering the consequences of their actions. The TRA theory also suggests that a person's intention to behave is predicted by their attitude towards that behavior and how they think others will judge them if they behave as such.

Oktaviani et al. (2017) stated that the tax compliance principle forces taxpayers to understand the laws and regulations regarding taxation so that they can carry out tax administration duties. Based on the Regulation of the Minister of Finance Number 192/PMK.03/2007, compliant taxpayers are taxpayers who are determined by the Director-General of Taxes as taxpayers who meet certain criteria. In measuring the Tax Compliant Intention variable, two approaches are needed: the tendency and the decision to comply. Concerning tax compliance, the intention is described as the Taxpayer's desire to perform tax compliance or non-compliance behavior. However, not all taxpayers' non-compliance is caused by the intention not to comply. The complexity of tax law also determines the occurrence of tax non-compliance in general in many places, so that tax non-compliance can occur due to non-intentional or unintended factors.

From all the literature above, we built the conceptual framework in Figure 1.

Research Hypothesis:

**Hypothesis 1 (H1).** *Religiosity has a significant effect on Attitudes.*

**Hypothesis 2 (H2).** *Religiosity has a significant effect on Tax Compliant Intention.*

**Hypothesis 3 (H3).** *Perceived Risk has a significant effect on Attitude.*

**Hypothesis 4 (H4).** *Perceived Risk has a significant effect on Tax Compliant Intention.*

**Hypothesis 5 (H5).** *Attitude has a significant effect on Tax Compliant Intention.*

**Hypothesis 6 (H6).** *e-Filing strengthens the effect of Religiosity on Tax Compliant Intention.*

**Hypothesis 7 (H7).** *e-Filing strengthens the effect of Perceived Risk on Tax Compliant Intention.*

**Hypothesis 8 (H8).** *e-Filing strengthens the effect of Attitudes on Tax Compliant Intention.*

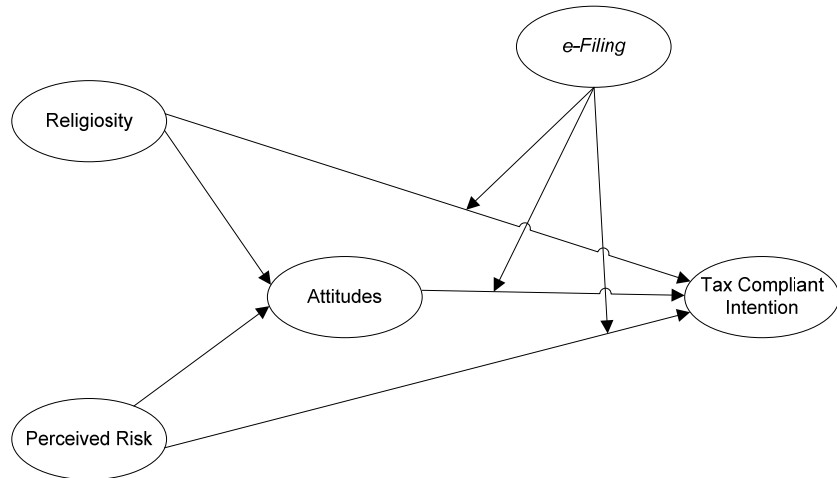

**Figure 1.** Conceptual Framework.

### 3. Research Methods and Materials

This research uses a quantitative approach, and the analysis is carried out on the numerical data from the measurement results of the variables. This study conducted a quantification study related to the phenomenon of policy implementation at the Directorate General of Taxation of Large Taxpayers Jakarta. The data were primary. Data collection involved the use of a questionnaire. This questionnaire design activity was carried out by selecting, adjusting and improving the statements of each indicator. The statement items that were selected and corrected were then combined with a Likert Scale model to produce a questionnaire. The Likert scale is a psychometric scale commonly used in questionnaires and is most widely used scale in survey research. The level of agreement referred to in this Likert scale consists of five scale choices that range from Strongly Agree (SS) to Strongly Disagree (STS).

Data analysis used the measurement model Structural Equation Model (SEM) with the help of the WarpPLS computer program/WarpPLS 6.0 software package, for the following reasons: (1). The analysis model is tiered and the structural equation model meets the recursive model. (2) Measurement of latent variables, namely any variables that cannot be measured directly.

In the SEM analysis, the design of the model is as follows:

Designing a structural model or inner model. The inner model states the relationship between latent variables (structural model). In the inner model, there are constant parameters. Latent and indicator variables (manifest variables) can be standardized without losing their generality. Therefore, the constant parameter is not presented in the model. The inner model equation is presented in the following equation:

$$Y = Y\beta + X\gamma + e$$

*Y*:    endogenous latent variable matrix
*X*:    exogenous latent variable matrix
β:    matrix of path coefficients of endogenous latent variables to endogenous latent variables
γ:    matrix of path coefficients of exogenous latent variables to endogenous latent variables
*e*:    inner model error matrix

Designing a measurement model or outer model. The outer model is the relationship between latent variables with indicators that are reflective or formative. The selection of an inappropriate model can lead to incorrect analysis results (Solimun et al. 2017). The basis for determining a variable is a reflective or a formative model based on theory, research rationale, and previous empirical research or empirical conditions. In this study, the indicator model used in the outer model is a reflective indicator.

The equation model can be written as follows:

$$x = \lambda_x X + u$$
$$y = \lambda_y Y + v$$

$y$:     indicator matrix for endogenous latent variables
$x$:     indicator matrix for exogenous latent variables
$Y$:     matrix for endogenous latent variables
$X$:     matrix for exogenous latent variables
$\lambda_y$:     loading matrix for endogenous latent variables
$\lambda_x$:     loading matrix for exogenous latent variables
$v$:     error for endogenous latent variable
$u$:     error for exogenous latent variable

Large taxpayers are generally in the form of agencies and individuals, so the population of this study was comprised of companies that are in the Directorate General of Taxation of Large Taxpayers Jakarta, Large Tax Service Offices 1 and 2, totaling 529 companies.

The sample units were chosen, namely Category One and Two Large Taxpayers, because they are relatively free private companies. Meanwhile, Category Three and Four Large Taxpayers include BUMN companies that have been regulated by law, so that they are relatively more obedient in paying taxes. Large One Taxpayer KPP, for certain Large Taxpayers who carry out business activities in the mining sector, mining support services, and financial services. Large Taxpayer Two Taxpayers, for certain Large Taxpayers who carry out business activities in the industrial, trade, and service sectors other than mining support services and financial services, for example, manufacturing, wholesale and retail trade, repair and maintenance of cars and motorbikes, information and communications, provision of electricity, gas, steam/hot water, and cold air, and installations. The sampling technique used in this study was simple random sampling. Thus, the sample size in this study was 155 large taxpayers (agencies) in Jakarta.

## 4. Results and Discussion

This study uses a research instrument in the form of a questionnaire. The questionnaire needs to be evaluated quantitatively, namely by checking the validity and reliability of the study. The results of the research instrument validity test are presented in Table 1.

Table 1 above shows the value of the corrected item correlation on the questionnaire for all indicators and items with a value above 0.3 (Solimun et al. 2017). Therefore, it can be concluded that all items have met the validity. The next stage is instrument reliability testing. The instrument is declared reliable if the Alpha-Cronbach value is >0.7 (Solimun et al. 2017). The results of the research instrument reliability test are presented in Table 2.

Table 2 shows that the Cronbach's Alpha value of the four research variables is more than 0.6. From these results, it can be concluded that Religiosity (X1), Perceived Risk (X2), e-Filing (M1), Attitude (Y1), and Tax Compliant Intention (Y2) are valid and reliable, so the data collected through this questionnaire can be used for data analysis at a later stage.

**Table 1.** Validity Test.

| Variable | Indicator | Correlation | Result |
|---|---|---|---|
| Religiosity (X1) | Faith (ideological) (X1.1) | 0.752 | Valid |
| | Religious Practices (X1.2) | 0.753 | Valid |
| | Appreciation (X1.3) | 0.789 | Valid |
| | Knowledge (intellectual) (X1.4) | 0.667 | Valid |
| | Practice (Consequences) (X1.5) | 0.827 | Valid |
| Perceived Risk (X2) | Financial Risk (X2.1) | 0.780 | Valid |
| | Functional Risk (X2.2) | 0.835 | Valid |
| | Physical Risk (X2.3) | 0.782 | Valid |
| | Psychological Risk (X2.4) | 0.807 | Valid |
| | Social Risks (X2.5) | 0.696 | Valid |
| | Time Risk (X2.6) | 0.748 | Valid |
| e-Filing (M1) | Frequency of Use (M1.1) | 0.775 | Valid |
| | The simplicity of the System (M1.2) | 0.760 | Valid |
| | Comprehensive Security (M1.3) | 0.725 | Valid |
| Attitude (Y1) | Behavioral Belief (Y1.1) | 0.735 | Valid |
| | Evaluation of Behavioral Belief (Y1.2) | 0.799 | Valid |
| Tax Compliant Intention (Y2) | Personal Tendency to Behave (Y3.1) | 0.676 | Valid |
| | The decision to Be Compliant (Y3.2) | 0.872 | Valid |

Notes: If Correlation Value ≥ 0.3, then the indicator is valid. Source: Research Data (2020).

**Table 2.** Reliability test.

| Variable | Alpha-Cronbach | Conclusion |
|---|---|---|
| Religiosity (X1) | 0.792 | Reliable |
| Perceived Risk (X2) | 0.754 | Reliable |
| *e-Filing* (M1) | 0.755 | Reliable |
| Attitude (Y1) | 0.751 | Reliable |
| Tax Compliant Intention (Y2) | 0.756 | Reliable |

Notes: If Alpha-Cronbach ≥ 0.7, then the variable is reliable. Source: Research Data (2020).

The first stage in WarpPLS research is measuring the outer model. There are two external measurements of the WarpPLS, namely the reflective and formative models. Table 2, shows the measurement model, the measurement weight value, and the p-value of each indicator for each variable.

Based on Table 3, it can be concluded that all of these latent variables have good and feasible indicators. Furthermore, this can be used to determine the most dominant indicator in contributing to the latent construct. The best indicator that forms Religiosity (X1) is belief (ideological) (X1.1), which has the highest factor loading of 0.291. The dominant indicator that forms Perceived Risk (X2) is Financial Risk (X2.1) with the highest loading factor of 0.266. Then, the dominant indicators that form e-Filing (M1) are the Frequency of Use (M1.1) and Comprehensive Security (M1.3), which have the highest loading factor of 0.410. The indicators that best form Attitude (Y1) are Behavioural Belief (Y1.1) and Evaluation of Behavioural Belief (Y1.2), which have the highest factor loading value of 0.559. Tax Compliant Intention (Y2) has an indicator with the highest loading factor of 0.558, namely Personal Tendency to Behaviour (Y2.1) and Decision to Be Compliant (Y2.2).

**Table 3.** Measurement model.

| Variable | Indicator | Weight | *p*-Value |
|---|---|---|---|
| Religiosity (X1) | Faith (ideological) (X1.1) | 0.291 | <0.001 |
| | Religious Practices (X1.2) | 0.284 | <0.001 |
| | Appreciation (X1.3) | 0.259 | <0.001 |
| | Knowledge (intellectual) (X1.4) | 0.262 | <0.001 |
| | Practice (Consequences) (X1.5) | 0.255 | <0.001 |
| Perceived Risk (X2) | Financial Risk (X2.1) | 0.266 | <0.001 |
| | Functional Risk (X2.2) | 0.245 | <0.001 |
| | Physical Risk (X2.3) | 0.246 | <0.001 |
| | Psychological Risk (X2.4) | 0.250 | <0.001 |
| | Social Risks (X2.5) | 0.251 | <0.001 |
| | Time Risk (X2.6) | 0.233 | <0.001 |
| e-Filing (M1) | Frequency of Use (M1.1) | 0.410 | <0.001 |
| | Simplicity of the System (M1.2) | 0.399 | <0.001 |
| | Comprehensive Security (M1.3) | 0.410 | <0.001 |
| Attitude (Y1) | Behavioral Belief (Y1.1) | 0.559 | <0.001 |
| | Evaluation of Behavioral Belief (Y1.2) | 0.559 | <0.001 |
| Tax Compliant Intention (Y2) | Personal Tendency to Behave (Y3.1) | 0.558 | <0.001 |
| | Decision to Be Compliant (Y3.2) | 0.558 | <0.001 |

Notes: If *p* value < 0.05, then the relationship is significant. Source: Research Data (2020).

The second stage in the WarpPLS research is to measure the inner model, also known as the structural model. The structural model presents the relationship between the research variables. The structural model coefficient explains the magnitude of the relationship between one variable and another. There is a significant effect between one variable on the other variable if the *p*-value < 0.05. In WarpPLS, there are two effects, namely the direct effect and the indirect effect. Table 4 presents the result for the direct effect test and Table 5 presents the result for the indirect effect test.

**Table 4.** Result of estimation and testing the direct effect.

| Relations between Variables | Hypothesis | Path Coefficient | *p*-Value | Conclusion |
|---|---|---|---|---|
| Religiosity → attitude | H1 | 0.239 | 0.001 | Significant |
| Religiosity → tax compliant intention | H2 | 0.255 | <0.001 | Significant |
| perceived risk → attitude | H3 | −0.207 | 0.004 | Significant |
| perceived risk → tax compliant intention | H4 | −0.161 | 0.020 | Significant |
| attitude → tax compliant intention | H5 | 0.186 | 0.008 | Significant |

Notes: If *p* value < 0.05, then the relationship is significant. Source: Research Data (2020).

**Table 5.** Result of estimation and testing of indirect effects.

| Indirect Effect | | | Coefficient | *p*-Value | Conclusion |
|---|---|---|---|---|---|
| Independent | → | Dependent | | | |
| Religiosity (X1) | → | Tax Compliant Intention (Y2) | 0.044 | 0.215 | Not significant |
| Perceived Risk (X2) | → | Tax Compliant Intention (Y2) | −0.039 | 0.247 | Not significant |

Notes: If *p* value < 0.05, then the relationship is significant. Source: Research Data (2020).

Table 4 presents the results of the following inner model testing:

(1)     The direct effect of religiosity on attitude has a path coefficient of 0.239 and a *p*-value of 0.001 (less than 0.05). There is a significant direct effect of religiosity on attitude. Considering that the path coefficient is positive, it can be concluded that when the religiosity increases, the attitude will also increase.

(2)     The direct effect of religiosity on tax compliant intention has a path coefficient of 0.255 and *p*-value < 0.001 (less than 0.05). There is a significant direct effect of religiosity on tax-compliant intention. Considering that the path coefficient is positive, it can be concluded that the higher the religiosity, the higher the intention to comply with taxes.

(3)     The direct effect of perceived risk on attitude has a path coefficient of −0.207 and a *p*-value of 0.004 (less than 0.05). There is a significant direct effect of perceived risk on attitude. Considering that the path coefficient is negative, it can be concluded that the higher the perceived risk, the lower the attitude will be.

(4)     The direct effect of perceived risk on tax compliant intention has a path coefficient of −0.161 and a *p*-value of 0.020 (less than 0.05). There is a significant direct effect of perceived risk on tax compliant intention. Considering that the path coefficient is negative, it can be concluded that the higher the perceived risk, the lower the tax compliant intention will be.

(5)     The direct effect of attitude on tax compliant intention has a path coefficient of 0.186 and a *p*-value of 0.008 (less than 0.05). There is a significant direct effect of attitude on tax compliant intention. Considering that the path coefficient is positive, it can be concluded that the higher the altitude, the higher the intention to comply with taxes will be.

The indirect effect of religiosity on tax compliant intention. Based on Table 5, religiosity (X1) has a positive but insignificant effect on tax compliant intention (Y2) with attitude (Y1) as mediation. This means that the attitude (Y1) is unable to mediate the relationship between religiosity (X1) and tax compliant intention (Y2) of 0.044 with a *p*-value of 0.215.

The indirect effect of perceived risk on tax compliant intention. Based on Table 5, perceived risk (X2) has a negative but insignificant effect on tax compliant intention (Y2) with attitude (Y1) as mediation. This means that the attitude (Y1) is unable to mediate the relationship between perceived risk (X2) and tax intention (Y2) of -0.039 with a *p*-value of 0.247.

*4.1. Moderation Effect of e-Filing on the Relationship between Religiosity on Tax Compliant Intention*

The SEM-WarpPLS analysis results obtained a moderation coefficient of 0.067 and a *p*-value of 0.199. Since the *p*-value is >0.05, e-Filing is not a moderation variable on the relationship between religiosity and tax compliant intention. Thus, there is only a significant positive direct effect on the relationship between religiosity and tax compliant intention. This shows that increasing religiosity can directly encourage an increase in the tax compliance intentions of large taxpayers' category one and two in Jakarta. The moderation effect of e-Filing on the Relationship between Religiosity on Tax Compliant Intention graphically can be seen in Figure 2.

*4.2. Moderation Effect of e-Filing on the Relationship between Perceived Risks on Tax Compliant Intention*

The SEM-WarpPLS analysis results obtained a moderation coefficient of −0.003 and a *p*-value of 0.354. Since the *p*-value is >0.05, e-Filing is not a moderation variable on the relationship between perceived risk on tax compliant intention. Thus, there is only a significant positive direct effect on the relationship between perceived risk and tax compliant intention. This shows that increasing perceived risk can directly encourage an increase in the tax compliance intentions of large taxpayers' category one and two in

Jakarta. The moderation effect of e-Filing on the Relationship between Perceived Risks on Tax Compliant Intention graphically can be seen in Figure 3.

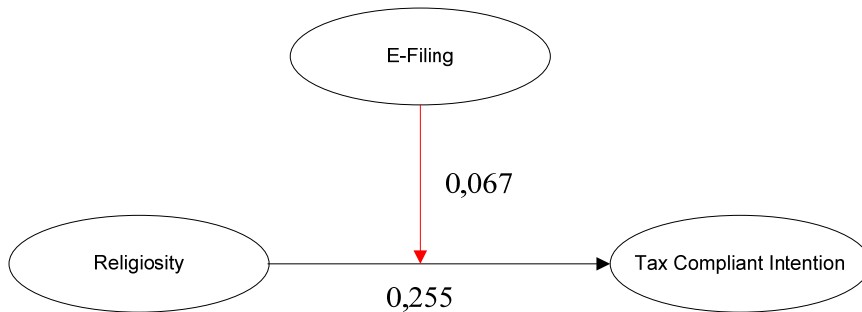

**Figure 2.** Moderation Effect of e-Filing on the Relationship between Religiosity on Tax Compliant Intention. Source: Research Data (2020).

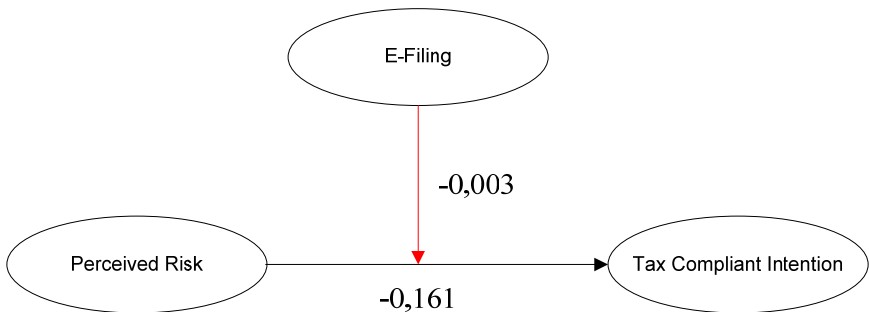

**Figure 3.** Moderation Effect of e-Filing on the Relationship between Perceived Risks on Tax Compliant Intention. Source: Research Data (2020).

*4.3. Moderation Effect of e-Filing on the Relationship between Attitudes on Tax Compliant Intention*

The SEM-WarpPLS analysis results obtained a moderation coefficient of 0.152 and a *p*-value of 0.026. Since the *p*-value is <0.05, e-Filing is a moderation variable on the relationship between attitude on tax compliant intention. This shows that e-Filing strengthens the relationship between attitude and tax compliance intentions of large taxpayers' category one and two in Jakarta. The moderation effect of e-Filing on the Relationship between Attitudes on Tax Compliant Intention graphically can be seen in Figure 4.

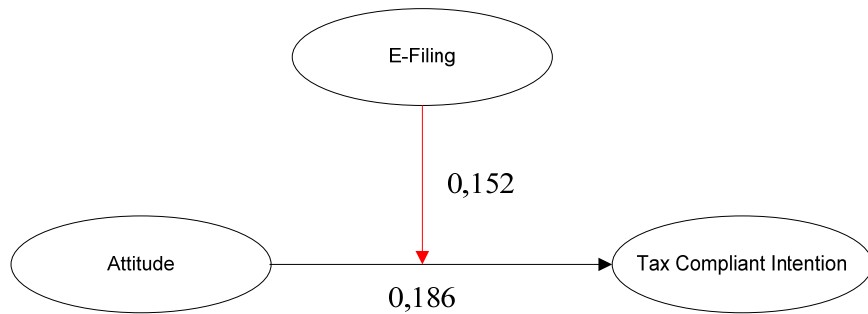

**Figure 4.** Moderation Effect of e-Filing on the Relationship between Attitudes on Tax Compliant Intention. Source: Research Data (2020).

Based on the Table 6, it can be seen that Religiosity (X1) is known to have a positive and significant effect on Attitude (Y1) and Perceived Risk (X2) is known to have a negative and significant effect on Attitude (Y1). Several recent studies have shown the importance of religious values in human resource management and organizational behavior in an organization. Several studies suggest that organizational change can be carried out through

the application of religious values in the workplace (Konz and Ryan 1999; Féry 2003; French and Bell 2001). One of the important aspects of making a change in the organization is the religious values that employees have. Several studies indicate that religiosity has a positive effect on work attitudes (McClelland 1961; Simmons and Parsons 2005; Weaver and Agle 2002). Indonesia, as a country that upholds religious values, should have a very strong ethical or moral foundation to avoid behavior that is detrimental to society at large. Various empirical studies have shown that the implementation of religiosity values can play a role in organizational change.

**Table 6.** Research Model Quality Index.

| Quality Index | Criteria | Statistics Value | Conclusion |
|---|---|---|---|
| Average path coefficient (APC) | Significant if $p < 0.05$ | 0.162 ($p = 0.010$) | Significant |
| Average R-squared (ARS) | Significant if $p < 0.05$ | 0.137 ($p = 0.020$) | Significant |
| Average adjusted R-squared (AARS) | Significant if $p < 0.05$ | 0.115 ($p = 0.036$) | Significant |
| Average block VIF (AVIF) | Accept if $\leq 5$ Ideal if $\leq 3.3$ | 1.101 | Ideal |
| Average full collinearity VIF (AFVIF) | Accept if $\leq 5$ Ideal if $\leq 3.3$ | 1.110 | Ideal |
| Tenenhaus GoF (GoF) | Small $\geq 0.1$ Medium $\geq 0.25$ Large $\geq 0.36$ | 0.277 | Medium |
| Sympson's paradox ratio (SPR) | Accept if $\geq 0.7$ Ideal if $= 1$ | 1.000 | Ideal |
| R-squared contribution ratio (RSCR) | Accept if $\geq 0.9$ Ideal if $= 1$ | 1.000 | Ideal |
| Statistical suppression ratio (SSR) | Accept if $\geq 0.7$ | 1.000 | Accept |
| Nonlinear bivariate causality direction ratio (NLBCDR) | Accept if $\geq 0.7$ | 1.000 | Accept |

Notes: The statistics value need to be adjusted to the existing criteria, if it exceeds the criteria then it is better. Source: Research Data (2020).

Religiosity is generally described as being related to cognition (religious knowledge, religious beliefs) which affects what is done with emotional attachment or emotional feelings about religion and/or behavior (Elci 2007). Religiosity according to Johnson et al. (2001) is seen as the extent to which individuals are committed to their religion and faith and apply their teachings so that individual attitudes and behaviors reflect this commitment. Worthington et al. (2010) call religiosity or religious commitment "the degree to which a person adheres to his/her religious values, beliefs, and practices, and uses them in daily living". Religiosity or religious commitment is divided into two types of commitments, namely intrapersonal religion, which comes from individual beliefs and attitudes, and interpersonal religious commitment, which comes from individual involvement with a community or religious organization. Mohd Ali (2013) states that the role of religious values is expected to spur positive behavior and prevent negative behavior towards behavioral compliance to encourage good behavior. Religiosity can be a factor that strengthens self-control from individuals and takes a positive role in preventing deviant behavior (Purnamasari and Amaliah 2015).

Today, understanding taxpayer compliance behavior is not easy. The biggest problem is when looking for the main theory of predictable taxpayer compliance, which is expected to produce voluntary compliance by taxpayers in a dynamic environment (Yusoff and Mohd 2017). This is a problem faced by the Directorate General of Taxes, whereby when

the functions of guidance or counseling, services, and supervision are running well, but the level of compliance with paying taxes does not work well with taxpayers. Many factors contribute to tax non-compliance and it is not easy to create perfect tax compliance.

This shows that Attitude (Y1) can be improved by first increasing Religiosity (X1) and decreasing the Perceived Risk (X2). Ajzen (1991) states that three independent factors determine a person's intention to behave. The first is the attitude towards behavior that leads to feelings of favorableness or unfavorableness towards an object that will be addressed, arising from an individual evaluation of the belief in the results obtained from the behavior.

However, Religiosity (X1) has a positive and significant effect on Tax Compliant Intention (Y2) through Attitude (Y1) and Perceived Risk (X2) has a negative and significant effect on Tax Compliant Intention (Y2) through Attitude (Y1). Individual behavior is based on the intention to behave (Lu and Wang 2018). The intention to behave is influenced by three beliefs, namely behavioral belief which forms attitudes, normative belief which forms subjective norms, and control belief which forms perceived behavioral control.

This means that an increase in Religiosity (X1) will lead to an increase in Attitude (Y1) and Perceived Risk (X2) will lead to a decrease in Attitude (Y1), which in turn will trigger an increase and decrease in Tax Compliant Intention (Y2). This shows that Attitude (Y1) has an important role of mediation between Religiosity (X1) and Perceived Risk (X2) toward Tax Compliant Intention (Y3). According to Ilyas and Burton (2013), for potential risks to be reduced organizations must implement risk management. The current condition of Indonesian taxation is required to implement risk management and reduce potential risks to a low level. Although final risk is in the form of financial risk, tax risk can be divided into four risks. First, operational risk, namely the risk caused by, among others, the inadequacy and/or malfunction of internal processes, human error, system error, or external problems that affect the operations of the Taxpayer. Second, legal risks, which are risks caused by weaknesses in juridical aspects, among others, due to legal claims from other parties, the absence of supporting statutory behavior or engagement weaknesses for a business transaction that has no specificity, clarity, or details regulating the transaction. Third, reputation risk, which is a risk caused by, among others, negative publicity related to business activities or negative perceptions of taxpayers. Fourth, compliance risk, which is the risk caused by taxpayers not complying with or implementing applicable tax laws and regulations.

Attitude has an important role in explaining a person's behavior in action, although many other factors influence behavior such as individual background and motivation. Reciprocally, environmental factors also influence attitudes and behavior. Attitude towards behavior is defined as an assessment of whether someone is favorable or not towards behavior. The higher the assessment, the greater the intention that is formed (Byabashaija and Katono 2011). Thus, an attitude shown by an individual can determine individual behavior (Night and Bananuka 2019; Nkundabanyanga et al. 2017; Chong et al. 2018). Economic growth raises the quality of life as people's income increases and directs their spending toward companies about which they have a positive attitude (Oh and Park 2020).

In addition, Attitude (Y1) is known to have a positive and significant effect on Tax Compliant Intention (Y2). This means that an increase in Attitude (Y1) will lead to an increase in Tax Compliant Intention (Y2). On the other hand, e-Filing has a negative and significant effect in strengthening the relationship between Attitude and Tax Compliant Intention. According to Tran-Nam (2015), there are three main models of tax compliance, namely prevention, fiscal psychology, and behavioral economics models. Traditional prevention models usually emphasize involuntary compliance through law enforcement such as audits, punishments, and prosecution. Fiscal psychological models emphasize voluntary compliance through preventive measures and education. Meanwhile, the behavioral economic model emphasizes compliance through the study of taxpayer behavior. Mainly considering education and study of taxpayer behavior, the state then launched an e-Filing initiative to make it easier for taxpayers to report tax. This will automatically be able to encourage an increase in the tax ratio and the achievement of tax revenues for the State.

*4.4. Research Limitations*

In this study, there are two limitations that cannot be provided by this study, namely:

1. Taxes are influenced by the variables of religiosity, e-Filing, perceived risk, attitudes, subjective norms, and perceptions of behavioral control. In future research, it is necessary to use variables that may affect tax compliance, such as tax avoidance, tax evasion, tax amnesty, trusted authorities, nationalism, and legal certainty. In addition, it is necessary to include variables that can represent the ability of the taxpayer, for example, financial statements, and leverage.

2. Considering that it is possible for the respondents of this study to have different points of view and assessments of the research variables, it is suggested that future research search for homogeneous respondents and broaden the scope. It is also recommended to use secondary data to minimize the possibility of bias in the respondents' perceptions obtained.

*4.5. Research Implications*

1. Improvements and improvements are needed regarding the use of e-Filing. The use of e-Filing can be improved in terms of the ease of use of the system so that it can be easily understood by taxpayers. The government in this case needs to improve and improve the previously existing features in e-Filing to make it more user friendly, simple, and systematic.

2. It is necessary to increase the religiosity element for the taxpayer community, considering that the results of this study state that religiosity can increase tax compliance attitudes and intentions. The government, through the Directorate General of Taxes under the coordination of the Ministry of Finance of the Republic of Indonesia, is advised to start incorporating elements of religion in conducting socialization related to tax compliance behavior or material related to taxation in terms of various religions in Indonesia.

3. The public as taxpayers are advised to support and encourage increased religiosity, attitudes, perceived risks, and the use of e-Filing in paying taxes, which has implications for tax compliance intentions where the use of tax revenues will be returned entirely for the prosperity of the Indonesian people.

4. The government through the Directorate General of Taxes under the coordination of the Ministry of Finance of the Republic of Indonesia is advised to measure the effectiveness of e-Filing performance in order to increase the level of taxpayer compliance. Increased taxpayer compliance will be able to assist the government in achieving tax revenue targets during the current year.

5. The government, through the Directorate General of Taxes under the coordination of the Ministry of Finance of the Republic of Indonesia is advised to provide an understanding of taxes using a religiosity approach, so that the attitude of taxpayers can be better and in order to increase the level of taxpayer compliance.

6. In future research, it is hoped several other variables will be combined that may become one of the driving and inhibiting factors in tax compliance intentions, for example tax avoidance, perceived behavior control, tax evasion, and others.

## 5. Conclusions

Based on the results of the empirical analysis, it can be concluded that there is a significant and positive effect between Religiosity and Attitude. There is a negative and significant effect between Perceived Risk on Attitude. There is a significant and positive effect between Attitude and Tax Compliant Intention. In addition, there is a positive but insignificant effect on the indirect relationship of Religiosity towards Tax Compliant Intention and a negative but insignificant effect on the indirect relationship of Perceived Risk towards Tax Compliant Intention. e-Filing strengthens the relationship between Attitude and Tax Compliant Intention.

Following the research results stated above, it can be argued that the results of this study can reveal a more comprehensive effect than previous studies regarding religiosity, attitudes, perceived risk, and e-Filing on tax compliant intention.

This research, both academically and practically, has several contributions which are described as follows:

(1) Comprehensively examines the variables that may affect the Intention to Comply in Paying Taxes.
(2) Provides additional knowledge and information for the government through the Directorate General of Taxes that in order to increase the Obedience of Large Taxpayers, it is necessary to provide guidance on the Religiosity of Large Taxpayers which, in this study, is represented by the Director of Finance.
(3) Provides information to the public that is important to improve taxpayer compliance because the use of tax revenues will be returned entirely for the prosperity of the people.

**Author Contributions:** Conceptualization (All Authors); methodology (M.S.U.); software (M.S.U.); validation (All Authors); formal analysis (All Authors); investigation, (All Authors); resources (M.S.U.); data curation (M.S.U.); writing—original draft preparation (M.S.U.); writing—review and editing (All Authors).; visualization (M.S.U.); supervision (U.N., K.H., A.P.); project administration (M.S.U.); funding acquisition (No Funding). All authors have read and agreed to the published version of the manuscript.

**Funding:** This research received no external funding.

**Institutional Review Board Statement:** Not applicable.

**Informed Consent Statement:** Not applicable.

**Data Availability Statement:** The data presented in this study are available on request from the corresponding author. The data are not publicly available due to privacy.

**Acknowledgments:** The author wants to thank the three anonymous referees for their thoughtful comments and suggestions, which have dramatically contributed to the improvement of this paper.

**Conflicts of Interest:** The author declares no conflict of interest.

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
