# Peer review of "Effect of Religiosity, Perceived Risk, and Attitude on Tax Compliant Intention Moderated by e-Filing"

_ijfs, doi:10.3390/ijfs10010008_

Round 1
Reviewer 1 Report
The paper investigates the impact of religiosity, perceived risk, and attitude, moderated by E-filing, on tax compliance intention by using a survey study.
Overall the paper contributes to the behavioral tax research on tax compliance. The questions addressed are interesting, the research methodology is generally appropriate, and the findings are promising. Nevertheless, some issues need to be discussed to improve and strengthen the paper.
First, the authors cite literature to explain religiosity, perceived risk, and attitude. More literature about the impacts of these factors on tax compliance behaviour should be discussed. For example, several studies examine the impact of religiosity on tax avoidance: Boone, J. P., Khurana, I.K., and Raman, K.K. 2013. Religiosity and Tax Avoidance, Journal of American Taxation Association 35(1), 53-84.
Second, the authors may provide more details about their data collection and questionnaire. For example, the questions asked on the questionnaire are evaluated based on 7-score scale? what questions are used to measure the variable such as Faith?
Lastly, tax compliance is an important issue for policy makers, professionals, public, academics and management. The authors should discuss more about the social and practical implications of their findings.
Author Response
I've done the revision by adding to the mark that I highlighted in yellow. I hope this answer can answer the reviewer's question in the review process that has been carried out. I really hope this paper can be accepted and published soon. Thank you very much for the help.
Reviewer 2 Report
A well designed and written article. So I accepted in present form of the paper.Author Response
I've done the revision by adding to the mark that I highlighted in yellow. I hope this answer can answer the reviewer's question in the review process that has been carried out. I really hope this paper can be accepted and published soon. I am very grateful that your review is very kind to me. Thank you very much for the help.
Reviewer 3 Report
The topic of the article is very interesting in the current socioeconomic context. However, the methodology must be extended and explained with a mathematical model to better understand the direct and indirect effects. In addition, a causality test is needed to assess the direction of causality between the variables and a more exhaustive analysis of possible spurious variables.
For this, in my opinion the paper could be accepted with major changes.
Author Response
Revisions have been made by putting yellow highlights on the manuscript that I have corrected. Thank you very much.
Round 2
Reviewer 3 Report
In my review, it was requested to write the corresponding model to estimate, which would facilitate the reading of the paper and help to understand the diagrams in Figure2-Figure 5. In addition, this would give a more academic and formal format to the paper.
Author Response
Thank you for your advise. I already revise your review about the methodology. Please see the paragraph with green highlight.